# Using ODRL to represent access rights to Public Records at The National Archives (UK)

Robert Walpole[1], Alex Green[2]

[1]*Devexe Limited, Alphington, Exeter, EX2 8YS, United Kingdom*
[2]*The National Archives, Kew, Richmond, Surrey TW9 4DU, United Kingdom*

### Abstract

This paper discusses a prospective model for describing access rights to public records held at The National Archives (TNA) using the Open Digital Rights Language (ODRL). In particular, it describes the approach taken to work out what ODRL policies would be needed to describe record closure. Record closure is a a record access policy derived from UK Government legislation. This legislation has evolved over time and this has resulted in a range of closure definitions for individual records depending on when they were transferred to TNA. It describes a method for generating large numbers of ODRL policy variants in RDF Turtle syntax, based on information held in an RDBMS, as well as a technical mechanism for linking these policies to the millions of records to which they apply.

### Keywords

ODRL, RDF, Legislation, Public Records, The National Archives, Freedom of Information

## 1. Introduction

### 1.1. The National Archives

The National Archives (TNA) is the official archive for the UK government and for England and Wales. TNA's catalogue holds over 11 million records covering a thousand years of history from the Domesday Book to websites. Records can take many forms, including paper or parchment, photographs, spreadsheets, websites, social media posts, posters, maps and drawings.

### 1.2. Legal framework for access to public records

The 1958 Public Record Act[1] formalised roles and responsibilities in respect of the transfer of records from government departments to the Public Record Office (a predecessor to The National Archives) no later than 50 years from their date of creation. At this point the records would be made available to the public. In practice records were often transferred before the 50 years and were held at TNA as closed records for release once they had reached 50 years. This closure period was reduced for new transfers to 30 years by The Public Record Act 1967[2] and then removed completely by the Freedom of Information Act 2000[3] which came into operation on 1 January 2005.

The FOI Act provided a new statutory framework for the access to public records which gave the public the right to see any public record immediately, wherever it is held, unless it was subject to an exemption to the FOI Act. However, the date of transfer from government departments to TNA was not altered until 2010, when an amendment to the FOI Act brought this forward from 30 to 20 years. This resulted in the accelerated transfer of records between 2013 to 2022, during which period two years of records were transferred every year.

In practice then, most records are transferred as open by the time they are 20 years old but some are closed under one or more FOI exemptions. These records may contain sensitive or distressing personal information or could damage international relations or national security if released. Others could be closed because they were transferred with an understanding that confidentiality would be maintained.

*1st NeXt-Generation Data Governance workshop @ SEMANTiCS 2024, September 17-19, 2024, Amsterdam, Netherlands*
✉ rob.walpole@devexe.co.uk (R. Walpole); alex.green@nationalarchives.gov.uk (A. Green)
🌐 https://www.devexe.co.uk/ (R. Walpole); https://www.nationalarchives.gov.uk (A. Green)

In addition to FOI exceptions, some records over 20 years old are retained by the government department. These are retained for up to 10 years and are subject to specific criteria, for example continued business in order to refer to maps of mines for urban planning. After 10 years, the government department must reapply to retain the record for a further period.

Under the FOI Act, all of these records are listed on the online catalogue including those with closed descriptions as the public have the right to place an FOI enquiry requesting the record is opened.

All applications to close or retain records are submitted to the Advisory Council on National Records and Archives. This body is chaired by the Master of the Rolls and is composed of academics, researchers, archivists, former officials and MPs. The Advisory Council scrutinises the applications, and those it agrees are passed to the Secretary of State to request final approval.

It is also important to note that there are two aspects of a record to consider in the context of making them available. Firstly, is the record itself subject to the legislation described above? Secondly, is the supplementary descriptive metadata about the record, usually termed its "description" considered sensitive? It is possible for a record to be closed but for the record description to be open. To use a military service record example, the record description might say that this is a service record for Joe Bloggs, who served as a Lieutenant in the Royal Navy, and allow the general public to see that, but not to view the record itself, which may contain the confidential medical information of someone who is still alive. It is also possible that both the record and record description are closed, possibly because the description itself contains sensitive information, and so all we can say is that there is a record in an identified series but no more.

## 1.3. Project Omega

Starting in 2019, TNA worked on Project Omega[4] to envision a new Linked Data[5] catalogue management system (referred to as the Pan Archival Catalogue, or PAC for short) which could replace a range of existing cataloguing systems within TNA and act as a single source of truth for TNA's records. A number of these existing cataloguing systems are over twenty years old, including the core editorial system PROCAT, which was created when The National Archives was still known as the Public Record Office (hence PRO). PROCAT is made up of two web-based GUI applications: an editor and a viewer, and backed by a relational database called the Inventory List Database, or ILDB for short. PROCAT allows staff to manage the metadata around new and existing physical records (a separate system is already in place for digital records).

In Project Omega's re-envisioning, a graph-based model was chosen for the new catalogue system as it was considered to be the most suitable for modelling the complex, and sometimes evolving, relationships between archival records. In itself, this is not a novel idea and in fact the International Council of Archives have been developing a graph-based model since 2012. However this model, known as Records in Contexts[6], or RiC for short, lacked several key features which TNA required including immutability, versioning, chain of history/provenance and multiple narratives. Instead, TNA selected the Matterhorn RDF Data Model[7] to provide the starting point for the new Catalogue Data Model[8].

One of the key attractions of the Matterhorn model was that it is not an ontology, like RiC, but rather an approach, which encourages the reuse of existing vocabularies and the creation of data shapes (using SHACL[9]) to describe the data model. This approach also aligned with one of the key drivers behind the project for TNA, which is the potential it offers for people to reuse and integrate with TNA data published on the web. Making use of familiar vocabularies, such as Dublin Core[10] and FOAF[11], is expected to make this easier for such users.

## 1.4. Access and closure within PROCAT

PROCAT is not the authoritative source of information about record access rights within TNA. For this, there is a separate system called SAR (System for Access Regulation) which contains some of the information in ILDB as well as additional information about access restrictions. The data is replicated from SAR to ILDB on a daily basis.

While it was anticipated that, in time, the information in SAR would be migrated to the Pan-Archival Catalogue, SAR was outside of the initial scope of Project Omega.

The following section describes how access conditions and closure are represented within PROCAT. To understand the meaning of some of the terminology it is necessary to understand how the catalogue is currently structured. The catalogue is a hierarchical system that loosely follows the ICA's ISAD(G) archival standard. TNA's catalogue consists of the following seven levels: :

- **Department** - mandatory, the top level grouping of records relating to an individual government department, executive agency or other government body
- **Division** - optional, used when it is desirable to group series together
- **Series** - mandatory, a grouping of records with a common history and purpose
- **Subseries** - optional, used when it is desirable to create groupings within a series
- **Subsubseries** - optional, used when it is necessary to group records below subseries
- **Piece** - mandatory, provides information about an individual record or group of records
- **Item** - optional, provides information about an individual record when it is desirable to split a piece, perhaps because of its physical bulk or when some documents within a piece require different closure status

An example of a catalogue record is as follows: ADM 53/119009. Here, ADM is the department, which in this case is the Admiralty, 53 is a series within the Admiralty, in this case containing ships' logs, and 119009 is a piece, which in this case is the log book of the ship HMS Birmingham for May 1944.

### 1.4.1. Access

The department to subsubseries levels of the catalogue have what are referred to as *access conditions*, rather than *closure*. It should be noted that, to date, only piece and item levels of the catalogue have had ODRL policies designed for them. This is because catalogue levels above piece level were out-of-scope for the initial phase of the project, and also because TNA is moving away from a hierarchical model of record organisation towards a poly-hierarchical one. For this reason and for the remainder of this paper, we will talk exclusively about *closure*, as described in the next section.

### 1.4.2. Closure

All records at the piece and item levels of the catalogue have *closure* information which is made up of the following four elements:

**1) Closure type**, which is stored in ILDB as one of the following characters with meaning as given:

- **A** - Open on transfer (default for piece and item)
- **N** - Normal closure 30 / Normal closure before FOI Act (from January 2005)
- **C** - Closed, for review in
- **D** - Retained until
- **U** - Closed until
- **F** - Closed for
- **I** - Open immediately
- **V** - Closed while access is reviewed (for FOI purposes only)
- **W** - Reclosed in (for FOI purposes only)
- **R** - Retained by department
- **S** - Retained by department under section 3.4
- **T** - Temporarily retained by department
- **X** - Unknown/unspecified

**2) Closure code**, which is stored in ILDB as an integer, the value of which is dependent to a degree on the closure type, as shown:

- *Normal closure before FOI Act* - always 30
- *Open on transfer* - always 0
- *Open immediately* - always 0
- *Closed, for review in* - must be a year (yyyy)
- *Closed until* - must be a year (yyyy)
- *Closed for* - must be a number of years (nn)
- *Reclosed in* - always a year (yyyy)
- *Retained until* - always a year (yyyy)

**3) Record opening date**, which is the date a closed piece or item will be made available. This date field is mandatory if records in the unit of description are closed.

**4) Closure status**, which is stored as one of the following characters with the meaning as given:

- **O** - Open Document, Open Description
- **D** - Closed or Retained Document, Open Description
- **C** - Closed or Retained Document, Closed Description

By looking at the information in all four of these fields together it is possible to make a statement about the closure of the record.

To give an example, the vast majority of the records within ILDB have the *closure type* N (Normal Closure before FOI Act). In this case the *closure code* should be 30, meaning 30 years. If the *record opening date* is present, we can say whether the record is open or closed depending on whether the opening date is in the past, present or future. If the *record opening date* is not available (which is the case for the vast majority of records) we need to look at a non-closure field in ILDB called the *covering end date*. This indicates the final date that any document in the record group (e.g. piece or item) was created or amended. By adding the *closure code* to the *covering end date* we discover the expected record opening date. So if the *covering end date* was 1st January 1944 and the *closure code* is 30 the record should be open from the 1st January 1974. The *closure status* tells us whether the record description is open or closed and, with regard to the record itself (the document), the status should match the calculated status.

As can be seen from this example, the data held in ILDB is not completely normalised and there is potential for ambiguity to exist between these fields. For example, there is nothing in the database to prevent a record being marked as open but for there to be no opening date. It is also theoretically possible for a record to have an opening date in the past and for the closure status to indicate that the record is closed.

## 2. Why ODRL?

The choice of the Open Digital Rights Language (ODRL) for describing access rights was a straightforward one, as Project Omega had already made the decision to use an RDF[12] graph model for the catalogue and also to follow the Matterhorn approach of reusing existing vocabularies. It was therefore a question of finding a vocabulary to describe access rights.

ODRL stood out as an established, well documented and flexible language for describing access rights as well as being the subject of two W3C recommendations[13][14]. This was important because as a UK government organisation, TNA is guided by the Government Digital Services (GDS) Service Manual which requires the use of open standards wherever possible[15].

This meant the only question left to resolve was whether ODRL would actually work for describing the closure information held in the catalogue. Failing this, it would most likely be that either a new bespoke vocabulary would have to be created to describe access rights, and/or an entirely new and as

yet undefined system would need to be built to manage them. As neither of these alternatives were attractive due to the considerable amount of additional development and maintenance burden that they would incur, a determined effort was made to see if ODRL would work.

## 3. Describing closure in ODRL

The ODRL Information Model talks about Assets[16], these are identifiable resources, such as a data object, a digital file or a physical artefact. In the context of TNA, we identified two assets: the record *description* and the record *realisation* (document).

Record descriptions are either open or closed, based on the *closure status* field, whereas the record documents are closed or open with optional constraints.

ODRL supports policy inheritance, where a child policy inherits all of the rules of a parent policy. Because we want to avoid information from being released unintentionally or prematurely, it was decided to have a default policy of closure, meaning there was a default prohibition on the general public reading anything about or within the records. This meant it would be necessary to explicitly state, at the record level, that permission had been granted to read the record or record description.

In terms of the semantic meaning, it might seem that there is little difference between a record that is closed until a certain date or closed for a certain number of years, if it means that they were both opened on the same date. However the archival metadata itself forms part of the public record at TNA and so it was important to preserve this subtle difference in meaning within the policies, if only at a descriptive level.

Establishing how to represent closure as ODRL policies required an iterative process, involving the Project Omega development team working together with members of the Catalogue and Access Management teams at TNA.

There were two main threads to these efforts: the first was to learn from others how ODRL had been applied to real life scenarios, and RightsML was our main source of reference in this regard; and the second was to analyse the closure data held in ILDB and attempt to describe every variation found with an ODRL policy. The resulting policies should retain all of the semantic meaning, data and intention of the original closure information and, at the same time, be fully machine readable.

Up to now, TNA has relied on there being a human in the loop to look at the data and make a decision about whether a record can be opened. While there is no expectation that this will change in the near future, it is not hard to imagine that this will become an overwhelming task once large volumes of digital records are included. Removing ambiguity in the closure metadata and making the policies machine readable (and more so, computable) was therefore an important consideration.

### 3.1. Learning from RightsML

RightsML showed us that being able to express closure using natural language would be a helpful guide. The following is taken from the IPTC Developer Site[17]:

> A Rights Expression Language (REL) is a machine-readable language to convey rights associated with a piece of content.

> The idea is to be able to automatically answer the question "Can we use this content for this particular purpose?" Rights are permissions and restrictions on the use of a piece of content, granted by a rights holder to a user. The basic structure is Party A grants Party B the right to Action C with Item D under Condition E

So, in our use case we might say for example:

> The National Archives (the assigner) denies (prohibition) The Public (assignees) that a record description or realisation (asset) may be read (action)

### 3.2. Analysing and cleaning the data using decision trees

To assist in the process of defining policies, the development team queried the closure information in all of the records held in ILDB, and then classified them into the different forms of closure, depending on the properties they contained. We were able to identify six fundamentally different forms of closure, and from these created decision trees. When applied to each individual record, the decision trees led either to a specific policy, which everybody agreed upon, or to further review by the Catalogue and Access Management teams. These decision trees were revised multiple times following discussion, with those records which caused most discussion being evaluated in more detail to understand why they had the properties they did. An example decision tree is shown in figure 1.

This process also highlighted a small number of anomalies in the records, for example some records had no date or conflicting information. The reason for these anomalies were investigated and corrected at source (i.e. within the ILDB database).

After multiple iterations of this process, we reached a point where all anomalies were eliminated and every record had a path through the decision trees which led to a policy. It was agreed to repeat this exercise of querying the database up until all records were migrated from PROCAT to PAC, to ensure that no new anomalies had crept into the data. If any records appeared that could not find a policy they would need to be reviewed.

### 3.3. The Closure Policies

At the end of this process we emerged with eighteen distinct policy types, of which thirteen could be assigned to records and two to record descriptions. The remaining three policy types were for inheritance purposes only and not intended to be assigned directly. Of these eighteen policies, all had the `odrl:Policy` type and eleven also had the `odrl:Offer` type, as they set either a prohibition or permission. Those policies without an `odrl:Offer` type only refined the descriptive metadata. The policies are shown in figure 2.

The actual number of distinct policies required is much greater than this as there is tremendous variation in the dates when access can be granted, and each distinct date requires a different policy. Despite this, the number of policies needed is far less than the number of records and most policies can be reused for multiple records.

## 4. Generating ODRL policies

Once it had been established that all closure could be modelled in ODRL, and ODRL example policies had been created for each scenario, it was possible to generate the policies directly from the closure information in the ILDB database. The mechanism for achieving this was already well established as it was the mechanism that had been used to export and transform all the other record information from ILDB into RDF. The Project Omega team used a tool called Pentaho Data Integration (a.k.a. Pentaho Kettle) from Hitachi Vantara[18]. This tool provides a framework for building repeatable data transformation workflows. Workflows are built up from a series of steps which can be dragged and dropped into a graphical user interface, with minimal coding required. For example a SQL query step can be added which only requires a database connection to be specified and a SQL query to be written for the data to start to flow into the pipeline. Further steps can be added to manipulate and transform the data as required. While many steps are available out of the box, Pentaho also supports plugin functionality, and as no functionality existed for creating, manipulating, validating and serialising RDF data, we reused a plugin which the project team had previously created for this purpose[19] within our new ODRL policy pipeline.

While the ODRL pipeline could create policies for each record, it did not create the higher level policies which these policies would inherit from. As there were only a small number of these higher level policies, it was decided they would be created manually.

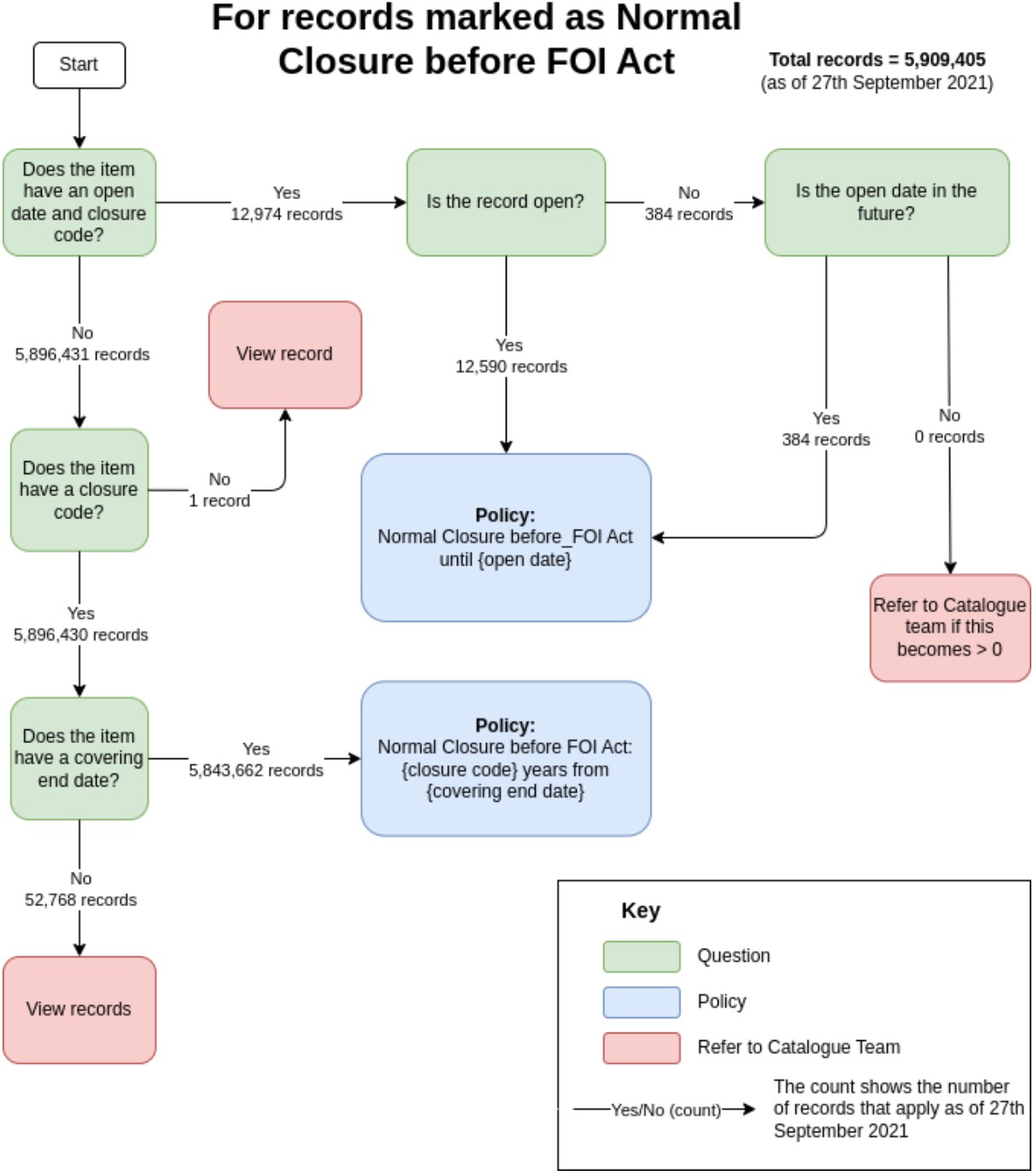

**Figure 1:** Example decision tree for assigning ODRL policies to records

The policy generation workflow can be seen in figure 3. There are two starting points within the workflow which use different SQL select statements, depending on the *closure type* (see 1.4.2). There are different paths through the workflow, depending on what closure information is present, but all paths lead either to the generation of a Apache Jena model[20], which is then validated and serialised to a Turtle RDF file, or to an error log. It is therefore essential to check the log files after the workflow has run to ensure that no errors were encountered. While errors do not prevent the workflow from completing, they are likely to mean that a policy has not been created for a particular closure scenario.

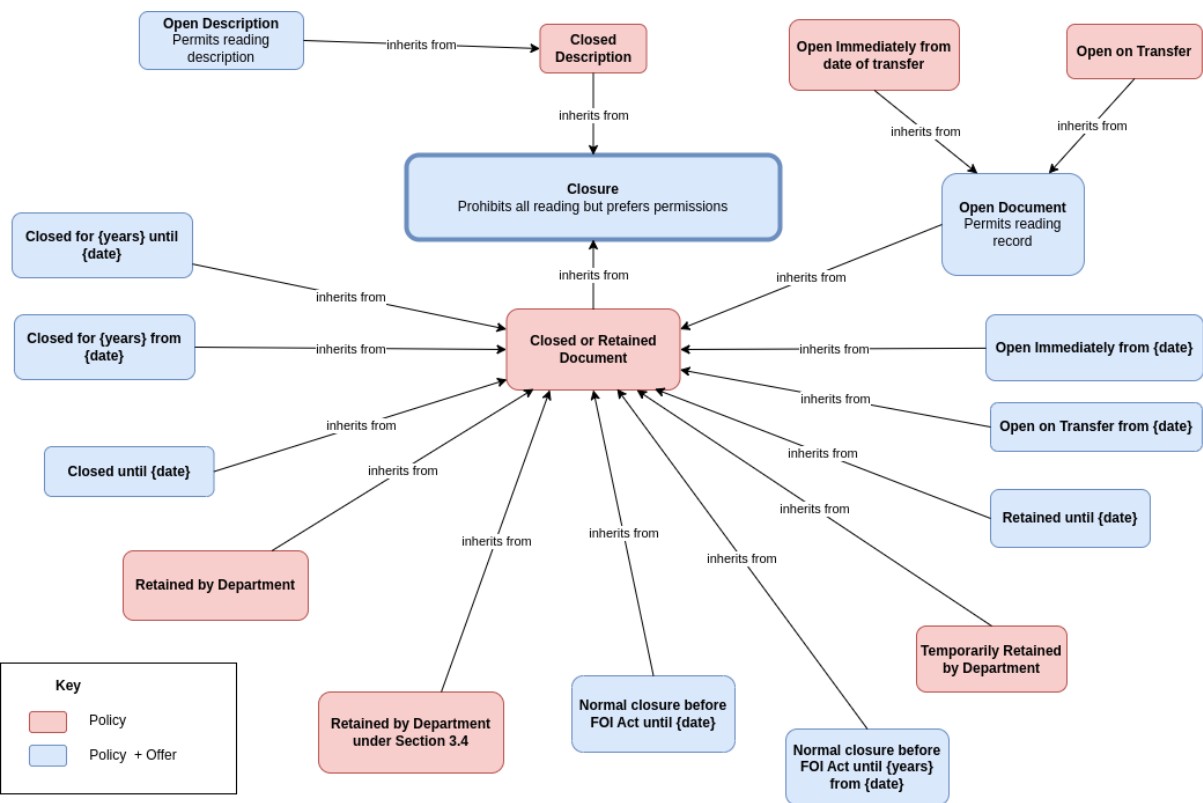

**Figure 2:** Diagram showing ODRL policy inheritance for closure

## 5. Assigning ODRL policies to records

Figure 4 shows a record with two policies: *Open Description* and *Open on Transfer*. Through inheritance, these policies grant permissions which override the prohibition on reading inherited from the parent policies of Closed Description and Closed or Retained Document, which in turn inherit from the Closure policy. This is dependent on the ancestor policy also setting the conflict strategy to prefer permissions.

As described previously, the *Closure* policy is the default policy from which all other policies inherit. The *Closure* policy is not intended to be applied directly to a record, rather policies which inherit from the *Closure* policy, and which add more descriptive and logical information should be applied. This requirement could be enforced either in the logic layer of the catalogue (i.e. if no policies exist for a resource then apply Closure) or at the storage layer (i.e. no resource can be stored without a valid policy), but how this is enforced is still to be decided. Most importantly, this inheritance ensures that access to a record has to be explicitly granted.

Policy assignment to records occurs within a sub-transformation of the record export job within Pentaho Kettle. In this sub-transformation (shown in figure 5) there are two User defined Java class steps which each allocate an ODRL policy URI, one for the record and the other for the record description. The selection of the URI occurs within the logic of the Java code inside the steps. These URIs are later linked to the record via the `odrl:hasPolicy` property. If policy assignment fails within this sub-transformation for any reason, an error will be logged and the policy will be assigned an invalid policy URI of either:

`http://catalogue.nationalarchives.gov.uk/policy.DESCRIPTION_POLICY_ERROR`

or:

`http://catalogue.nationalarchives.gov.uk/policy.DOCUMENT_POLICY_ERROR`

The intention of this is that the invalid URIs will be picked up during validation. At the same time, the export job is able to continue unhindered. This is important, as the export job can take several hours to

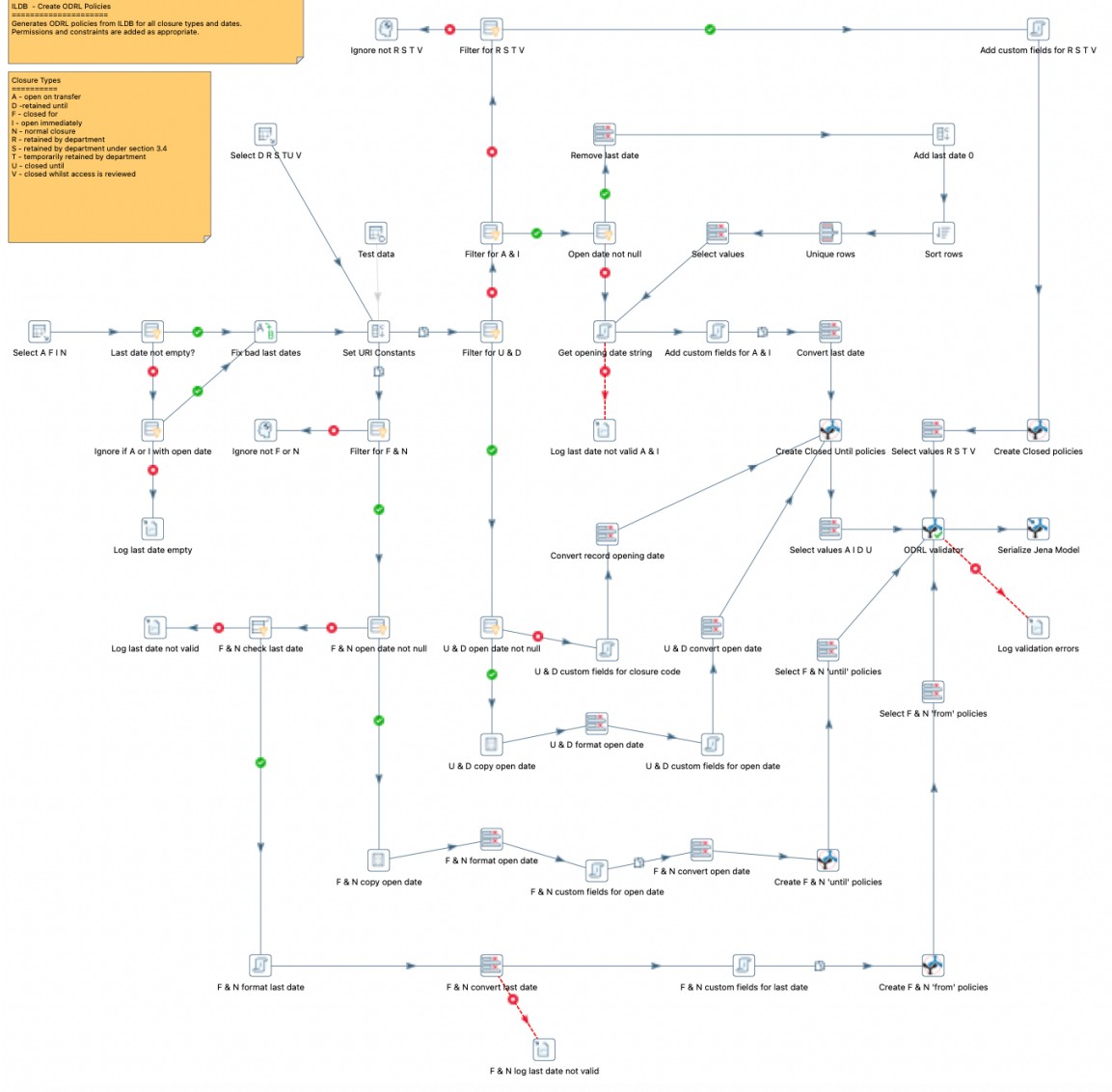

**Figure 3:** Policy generation transformation shown in Pentaho Kettle UI

run, and if an error causes it to halt, a great deal of time can be wasted. It is always preferable to allow the export to complete and then check for errors. Once the error is corrected the job can be re-run just for the specific records that caused errors on the initial run.

## 6. Validating ODRL policies and records using SHACL

Because the ODRL Information Model has been built using Linked Data[5] principles, it is fully compatible with the Open World Assumption[21], meaning that just because a statement doesn't exist about a resource, that doesn't mean it can't be true. It also means that having a statement about a resource doesn't preclude another, perhaps seemingly contradictory, statement being added. When it comes to validating that any policies and rules created for records adhere, not only to the ODRL Information Model, but also to the specific requirements of closure, this poses a challenge. For example, in the closure model, all closure policies which are assigned to records must inherit from one other policy (see figure 4) but there is nothing within RDF itself that can be used to enforce this. This leads to the possibility of mistakes being made which could remain undetected and ultimately lead to false conclusions about record closure being drawn.

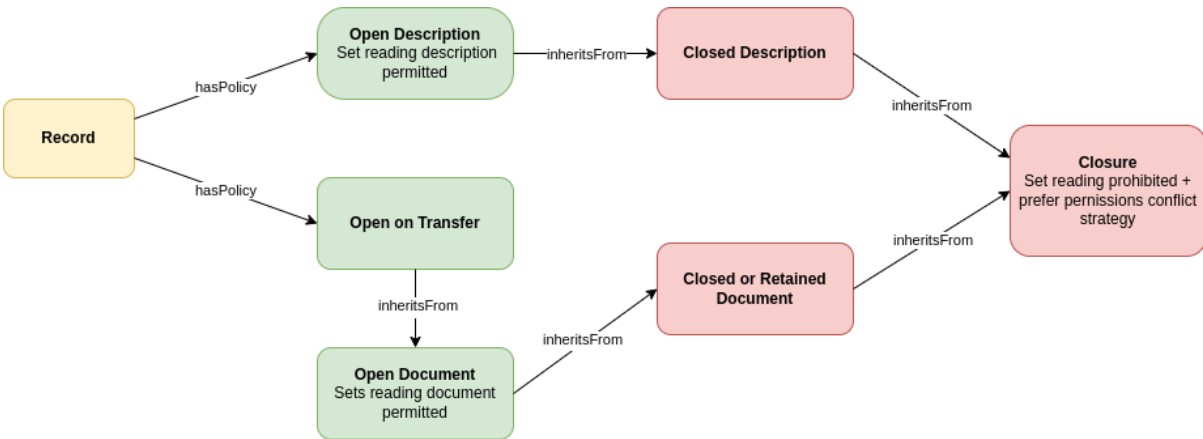

**Figure 4:** ODRL policy inheritance shown in relation to a record

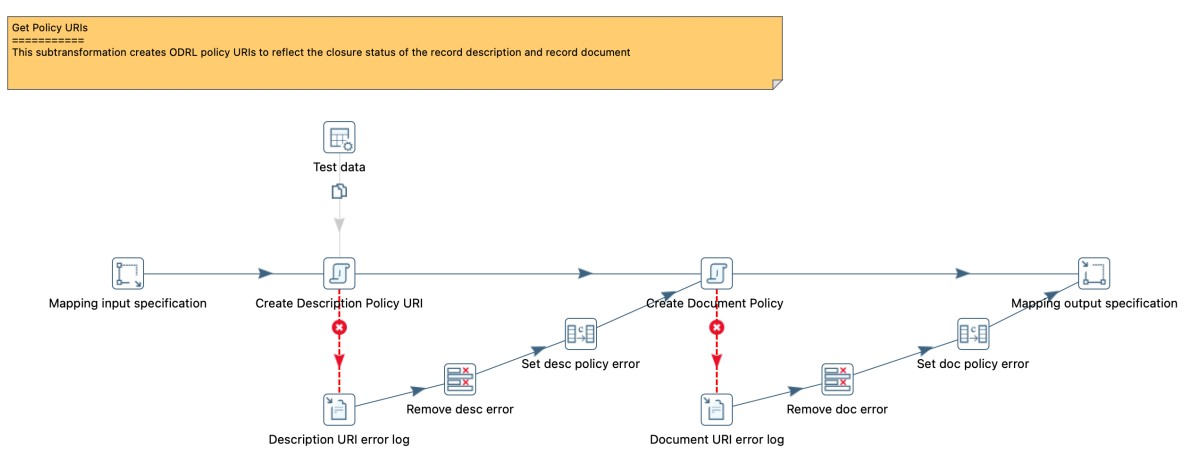

**Figure 5:** Policy assignment sub-transformation shown in Pentaho Kettle UI

One solution is provided by the Shapes Constraint Language (SHACL)[9] which allows expected data patterns or "shape graphs" written in RDF to be compared with the actual "data graphs" containing the policies and rules that have been created. The W3C Permissions and Obligations Expression Working Group provide some examples of SHACL shapes for validating policies at `https://www.w3.org/2016/poe/wiki/Validation#Validation`.

Following the approach shown in these examples, the Project Omega development team created custom SHACL shapes for closure which could be used to validate all the policies generated by the policy generation workflow. Furthermore we have incorporated SHACL validation into the workflow itself, as can be seen in the *ODRL validator* step in figure 3. This was achieved by incorporating Apache Jena SHACL[22] into the Kettle Jena Plugins[19] mentioned previously.

## 7. Is it open or closed?

When there is a need to know whether a record can be viewed by a member of the public, a logical evaluation must take place in which all of the allocated and inherited permissions and prohibitions for the record are merged and assessed. As the data is stored in RDF, a SPARQL query can be used for this purpose, but it would also be possible (and probably easier) to use an RDF API such as RDF4J or Apache Jena ARQ to build these queries. An example SPARQL query for this purpose is shown in Appendix A.

The result of this assessment is the dynamic creation of a custom ODRL policy for the specific record. This policy will contain only the discovered rules (permissions and prohibitions) of all of the relevant policies. For an example of such a policy see Appendix B.

### 7.1. Conflict strategy and ODRL profile

Unless the record and description are closed, there will be a conflict arising between these rules because the root policy of Closure states that reading is not allowed. Any permission granted to read a document or description will conflict with this. ODRL provides a conflict strategy mechanism[23] to deal with this which can result in either prohibitions overriding permissions or permissions overriding prohibitions. PAC would take the latter approach as this allows *Closure* to be the default position, with the catalogue team needing to make a positive choice to allow a user controlled access to the record or record collection. This will help to prevent accidental publishing of closed records.

The core ODRL profile defines a *Read* action to which permissions can be granted. It became apparent, while modelling the closure information in the catalogue, that it would be very helpful to distinguish the action of reading the description from the action of reading the record document. This way there would be no ambiguity about what action the permission was granting. The ODRL Profile Mechanism[24] allows direct extension of the ODRL core vocabulary with additional semantics. This allows additional ODRL actions for *Read Document* and *Read Description* to be created which extend the core *Read* action by incorporating an *Included In* relationship. The TNA ODRL profile is essentially a small OWL[25] ontology and all definitions are within the ODRL namespace. The additional terms created are therefore:

- `http://www.w3.org/ns/odrl/2/readDocument`
- `http://www.w3.org/ns/odrl/2/readDescription`

The draft ODRL profile file for The National Archives can be seen in Appendix C.

## 8. Conclusions and future work

The use of ODRL described in this paper, and the outcome of other design choices developed and tested during Project Omega, is currently the subject of a review by TNA while they decide on a future direction.

To date, only the closure information for piece and item levels of the catalogue has been resolved, meaning more work needs to be done to decide if and how ODRL policies would be used to describe access to higher levels of the catalogue.

There is also an open question about what happens when a closed record reaches a date specified in a constraint for opening. Does this mean that the record can be read by the public from this date, or does this mean that at that point, TNA will review the record, and it may or may not be opened as a result? In this case we are not talking about public access but about a business process which The National Archives needs to follow. It may be possible to model this process in ODRL by saying something like:

> The National Archives has a duty to review access to record A on the 1st January 2025

Additionally, there may be other novel ways to model this process outside of ODRL. This remains a research area of outstanding interest.

In general it is expected that ODRL will allow TNA to simplify how closure and access is applied to records and their descriptions in future. The process of applying ODRL policies to records and having to decide what prohibitions and permissions are needed has been extremely instructive in terms of better understanding what information is needed for a logical assessment of closure to be made. The large number of policy variations with varying descriptions but essentially identical effects are unlikely to be needed for future records. We now know that record closure can be defined very precisely in terms of prohibitions or permissions with optional constraints. At the same time, using a Linked Data[5] model will enable us to link our ODRL policies to specific clauses of the Public Records Act (and other

legislation) via the legislation published as RDF on Legislation.gov.uk. Linking policies directly to the appropriate legislation could even make the need for closure descriptions redundant.

ODRL's inherent flexibility means that in future it should also be possible to describe different degrees of access to records, above and beyond closure. For example, TNA might want to differentiate between a record which can be delivered over the Internet and one which requires a visit to the invigilation room at TNA for viewing. In another scenario, TNA might want to withhold specific records from Google search, but still make them publicly available through their online catalogue.

## 9. Acknowledgments

Thank you to Dr K Faith Lawrence and Dr Jenny Bunn at The National Archives for their identification of funding sources and other support.

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

# A.  Example SPARQL query to evaluate inherited permissions and prohibitions for a record

```
PREFIX cat: <http://cat.nationalarchives.gov.uk/>
PREFIX odrl: <http://www.w3.org/ns/odrl/2/>
PREFIX nat: <http://www.nationalarchives.gov.uk/>
PREFIX dct: <http://purl.org/dc/terms/>

BASE <http://cat.nationalarchives.gov.uk/>

CONSTRUCT {
    ?policyUri a odrl:Policy, odrl:Offer ;
        odrl:profile nat:odrl-profile ;
        odrl:conflict ?conflict ;
        odrl:permission ?b1 ;
        odrl:prohibition ?b2 .
    ?b1 ?p1 ?v1 ;
        odrl:assigner cat:The_National_Archives ;
        odrl:target ?resource .
    ?b2 ?p2 ?v2 ;
        odrl:assigner cat:The_National_Archives ;
        odrl:target ?resource .
    ?b1 odrl:constraint ?c1 .
    ?c1 ?c2 ?c3 .
}
WHERE
{
  BIND(cat:ADM.2021.21L1TH.P.1 AS ?resource)
  ?resource dct:identifier ?identifier
    BIND(URI(CONCAT(?identifier,"-policy")) AS ?policyUri)
    {
        {
            SELECT DISTINCT ?permission
            WHERE {
                BIND(cat:ADM.2021.21L1TH.P.1 AS ?resource)
```

```
            ?resource dct:accessRights ?accessRights .
            ?accessRights odrl:hasPolicy ?policy .
            { ?policy odrl:permission ?permission . }
            UNION
            {
                ?policy odrl:inheritFrom+ ?parentPolicy .
                ?parentPolicy odrl:permission ?permission
            }
        }
    }
    BIND(BNODE() AS ?b1)
    OPTIONAL {
        ?permission odrl:constraint ?constraint .
        BIND(BNODE() AS ?c1)
        ?constraint ?c2 ?c3 .
    }
    ?permission ?p1 ?v1 .
    FILTER(?p1 NOT IN(odrl:constraint)) .
}
UNION
{
    {
        SELECT DISTINCT ?prohibition
        WHERE {
            BIND(cat:ADM.2021.21L1TH.P.1 AS ?resource)
            ?resource dct:accessRights ?accessRights .
            ?accessRights odrl:hasPolicy ?policy .
            { ?policy odrl:prohibition ?prohibition . }
            UNION
            {
                ?policy odrl:inheritFrom+ ?parentPolicy .
                ?parentPolicy odrl:prohibition ?prohibition
            }
        }
    }
    BIND(BNODE() AS ?b2)
    ?prohibition ?p2 ?v2 .
}
{
    SELECT DISTINCT ?conflict
    WHERE {
        BIND(cat:ADM.2021.21L1TH.P.1 AS ?resource)
        ?resource dct:accessRights ?accessRights .
        ?accessRights odrl:hasPolicy ?policy .
        { ?policy odrl:conflict ?conflict . }
        UNION
        {
            ?policy odrl:inheritFrom+ ?parentPolicy .
            ?parentPolicy odrl:conflict ?conflict .
        }
    }
}
```

```
}
```

# B. Example of constructed policy for a record based on incorporating inherited permissions and prohibitions

The example shown represents a record with an open description and open document with effect from the 14th July 2015.

```
@prefix nat:  <http://www.nationalarchives.gov.uk/> .
@prefix rdf:  <http://www.w3.org/1999/02/22-rdf-syntax-ns#> .
@prefix cat:  <http://cat.nationalarchives.gov.uk/> .
@prefix xsd:  <http://www.w3.org/2001/XMLSchema#> .
@prefix odrl: <http://www.w3.org/ns/odrl/2/> .

cat:ADM.2021.21L1TH.P.1-policy
    rdf:type          odrl:Offer, odrl:Policy ;
    odrl:conflict     odrl:perm ;
    odrl:permission   [
        rdf:type          odrl:Permission ;
        odrl:action       odrl:readDocument ;
        odrl:assignee     cat:The_Public ;
        odrl:assigner     cat:The_National_Archives ;
        odrl:constraint [
            rdf:type          odrl:Constraint ;
            odrl:leftOperand  odrl:dateTime ;
            odrl:operator     odrl:gteq ;
            odrl:rightOperand "2015-07-14"^^xsd:date
        ] ;
        odrl:target       cat:ADM.2021.21L1TH.P.1
    ] ;
    odrl:permission   [
        rdf:type        odrl:Permission ;
        odrl:action     odrl:readDescription ;
        odrl:assignee   cat:The_Public ;
        odrl:assigner   cat:The_National_Archives ;
        odrl:target     cat:ADM.2021.21L1TH.P.1
    ] ;
    odrl:profile      nat:odrl-profile ;
    odrl:prohibition [
        rdf:type        odrl:Prohibition ;
        odrl:action     odrl:read ;
        odrl:assignee   cat:The_Public ;
        odrl:assigner   cat:The_National_Archives ;
        odrl:target     cat:ADM.2021.21L1TH.P.1
    ] .
```

# C. The National Archives ODRL profile (draft)

```
@prefix owl:  <http://www.w3.org/2002/07/owl#> .
@prefix rdf:  <http://www.w3.org/1999/02/22-rdf-syntax-ns#> .
@prefix odrl: <http://www.w3.org/ns/odrl/2/> .
```

```
@prefix rdfs: <http://www.w3.org/2000/01/rdf-schema#> .
@prefix skos: <http://www.w3.org/2004/02/skos/core#> .

<http://www.nationalarchives.gov.uk/odrl-profile>
    rdf:type owl:Ontology ;
    owl:imports odrl: ;
    rdfs:label "The National Archives ODRL Profile"@en .

#############################################################
#    Individuals
#############################################################

###  http://www.nationalarchives.gov.uk/odrl-profile/#actions
<http://www.nationalarchives.gov.uk/odrl-profile/#actions>
    rdf:type owl:NamedIndividual , skos:Collection ;
    skos:member odrl:readDescription ,
    odrl:readDocument ;
    skos:prefLabel "Actions for Rules"@en ;
    skos:scopeNote "The National Archives ODRL Profile"@en .

###  http://www.w3.org/ns/odrl/2/odrl-profile
odrl:odrl-profile
    rdf:type owl:NamedIndividual , skos:Collection ;
    skos:member odrl:readDescription , odrl:readDocument ;
    skos:prefLabel "The National Archives ODRL Profile"@en .

###  http://www.w3.org/ns/odrl/2/readDescription
odrl:readDescription
    rdf:type owl:NamedIndividual , skos:Concept , odrl:Action ;
    odrl:includedIn odrl:read ;
    rdfs:isDefinedBy odrl: ;
    rdfs:label "Read Description"@en ;
    skos:definition "To read a record description." .

###  http://www.w3.org/ns/odrl/2/readDocument
odrl:readDocument
    rdf:type owl:NamedIndividual , skos:Concept , odrl:Action ;
    odrl:includedIn odrl:read ;
    rdfs:isDefinedBy odrl: ;
    rdfs:label "Read Document"@en ;
    skos:definition "To read a record document." .
```