# OpenReview forum: "Using ODRL to represent access rights to Public Records at The National Archives (UK)"
_SEMANTiCS.cc/2024/Workshop/NXDG — NXDG 2024_

### Official Review · ~Patrick_Hochstenbach1 · 2024-08-01
**Implementing ODRL in The National Archives**

**Rating:** 7
**Confidence:** 4

**Review:**

The National Archives (TNA) catalogue holds 11 million archival records representing their collections. These records provide a hierarchical description of pieces and item artifacts available in the archive. The artifacts and metadata about the artifacts have various access conditions. Both artifacts and metadata can contain sensitive personal information or be connected to national security issues. Policies define when artifacts and/or metadata about the artifacts can be made available to a particular audience. TNA is moving to Linked Data to describe the collections. In a similar move, the description of access policies will be moved to ODRL for two reasons: 1) the flexibility of the extensibility data model and 2) the possibility of automating access decisions.

The paper provides an overview of the TNA's current policy model and how it can be mapped to the ODRL model. Next, it details how current policies are transformed into ODRL using Pentaho data transformation tools, validated using SHACL, and queried using SPARQL.

From a technical viewpoint, the most interesting part for the audience will be the discussion of conflict resolution in ODRL, where a hierarchy of policies will reach different conclusions about the accessibility level of the artifact (or its metadata). TNA requires an extension of the ODRL model to fit their local use cases. I hope to see more details about this part in the workshop.

Although it is clear that ODRL policies need to be validated to be trusted in a production environment, the motivation for this is not entirely convincing. The authors write that due to the Open World Assumption, "meaning that new information may come to light at any time, which could run counter to any conclusions drawn based on existing information," validation is required. This premise, without providing extra context as to why this is the case, is misleading. RDF logic is monotonic, meaning that any added information cannot possibly invalidate previous conclusions. To invalidate previous conclusions, something extra (that is not stated) is implied. Maybe there are added semantics in ODRL that create these types of risks? How SHACL can mitigate these risks is unclear. I guess SHACL is required for other reasons (e.g., verifying policy shapes against a mandatory shape). This is also related to conflict resolution, and I would assume the workshop is a good place to discuss this topic.

---

### Official Review · ~Víctor_Rodríguez-Doncel1 · 2024-08-01
**Good example of use of ODRL**

**Rating:** 8
**Confidence:** 5

**Review:**

This paper describes the use of the W3C specification for policies, Open Digital Rights Language (ODRL), in order to manage the access rights to public records at The National Archives (TNA).
This paper is not a research paper, and as such, it would be not much appreciated in other venues. But the paper is ideal for a venue like SEMANTICS, where practical applications, real cases, and successful implementations are welcome. Under this perspective the paper is excellent: it describes well the problem, the solution is sound and it includes real examples (SPARQL queries) and very illustrative diagrams.
The paper convincingly shows how ODRL can be used to encode a wide range of access policies in a machine-readable format, which is critical for modernizing and improving the accessibility and transparency of archival records management.
Those with intention to make a similar implementation will find the paper valuable.
Clear accept.

---

### Decision · Program_Chairs · 2024-08-02

Accept